# Relationship of the Aggregation of Cardiovascular Risk Factors in the Parasympathetic Modulation of Young People with Type 1 Diabetes

**DOI:** 10.3390/medicina55090534

**Published:** 2019-08-26

**Authors:** Anne Kastelianne França da Silva, Diego Giulliano Destro Christofaro, Laís Manata Vanzella, Franciele Marques Vanderlei, Maria Júlia Lopez Laurino, Luiz Carlos Marques Vanderlei

**Affiliations:** 1Department of Physical Therapy, Faculty of Science and Technology, São Paulo State University (UNESP), Roberto Simonsen Street, 305, Presidente Prudente, São Paulo 19060-900, Brazil; 2Department of Physical Education, Faculty of Science and Technology, São Paulo State University (UNESP), Roberto Simonsen Street, 305, Presidente Prudente, São Paulo 19060-900, Brazil

**Keywords:** autonomic nervous system, diabetes mellitus, type 1, risk factors, young adult

## Abstract

*Background and objectives:* In healthy individuals, autonomic alterations are associated with the aggregation of cardiovascular risk factors. However, in individuals with type 1 diabetes, who are known to present autonomic alterations, mainly characterized by a reduction in parasympathetic modulation, these associations have not yet been investigated. We assess whether the aggregation of cardiovascular risk factors influences parasympathetic indices of heart rate variability in young people with type 1 diabetes. *Materials and methods:* This cross-sectional study included 39 individuals with type 1 diabetes (22.54 ± 4.31), evaluated in relation to the risk factors: blood pressure, fat percentage, and resting heart rate. For heart rate variability analysis, heart rate was recorded beat-to-beat using a cardio frequency meter (PolarS810i) for 30 min with the volunteers in dorsal decubitus. The parasympathetic heart rate variability indices were calculated: rMSSD, pNN50, high frequency (HF) n.u (normalized units), SD1, 2LV, and 2ULV. Data collection was carried out in 2014 and analyzed in 2017. *Results:* Individuals with two aggregate risk factors present a reduction in the values of the indices that reflect parasympathetic autonomic modulation compared to individuals without the risk factors analyzed, regardless of sex and age. *Conclusion:* In young people with type 1 diabetes, the aggregation of cardiovascular risk factors is associated with parasympathetic autonomic impairment.

## 1. Introduction

Studies have pointed out that the aggregation of one or more risk factors is associated with a reduction in cardiac autonomic modulation in children [1], adolescents [2], and healthy adults [3]. An early indicator of alterations in autonomic modulation is a reduction in heart rate variability (HRV) [4,5]. The use of this indicator enables identification of an overall reduction in the HRV in individuals with type 1 diabetes, when compared with healthy individuals [6,7], as well as a reduction in both components of the autonomic nervous system (ANS) [8], or loss in the parasympathetic component with an increase in the sympathetic component [7]. A meta-analysis performed by Maser et al. [9] showed that a reduction in autonomic cardiovascular function is strongly associated with an increased risk of silent myocardial ischemia and mortality in individuals with diabetes. 

The presence of risk factors is associated with negative modifications in the autonomic behavior in individuals with type 1 diabetes. Colhoun et al. [10] studied 160 adults with type 1 diabetes, and found an inverse association between global variability and several other factors, such as age, disease duration, higher blood pressure, body mass index, waist/hip ratio, triglycerides, HbA 1c, and physical activity. Similar data were also provided by the EURODIAB Prospective Complications Study Group [11], who highlighted HbA1c, hypertension, distal symmetrical polyneuropathy, and retinopathy as factors capable of predicting the risk of cardiac autonomic dysfunction in type 1 diabetics in a period of 7.3 years. 

From the above, it is understood that the population with type 1 diabetes can demonstrate several complications, such as the development of cardiovascular disease (CVD) and changes in autonomic behavior, and, furthermore, that risk behaviors, when aggregated, may favor the development of risk factors for CVD even in healthy individuals. Modifiable risk factors, such as blood pressure, body fat, and heart rate, which can be easily measured in clinical practice, has shown an important association with autonomic control [12,13,14] and risk of CVD [15,16]. Also, these factors are less investigated in the diabetes populations, since most publications focus more on glucose management [15] than on other risks factors. Due to the importance of autonomic control for diabetes populations, the high number of risk factors for CVD, and the possibility of these factors being present in the same individual, the novelty of the present study is to evaluate the relationship between the aggregation of CVD risk factors (blood pressure, resting heart rate, and fat percentage) and parasympathetic indices of heart rate variability in young people with type 1 diabetes.

## 2. Methods

### 2.1. Population

This cross-sectional study followed the Strengthening the Reporting of Observational Studies in Epidemiology (STROBE) recommendations (Appendix A) [17] and included voluntary participation of 43 young individuals, aged between 18 and 30 years, of both sexes (20 men and 23 women), with a clinical diagnosis of type 1 diabetes (disease duration of 11.20 ± 6.01 years), recruited from a database of Basic Health Units and direct contact with doctors specialized in endocrinology, both in the city of Presidente Prudente, SP, Brazil. For the calculation of the sample, a value of r = 0.45 was used with an alpha error of 5% and statistical power of 80%, which generated a minimum number of 38 subjects to be evaluated in this study.

To be part of this study, volunteers were required to present a clinical diagnosis of type 1 diabetes confirmed by medical diagnosis, not be smokers and/or alcoholics, and not present known cardiorespiratory diseases (e.g., chronic obstructive pulmonary disease, ischemic coronary artery disease). The volunteers who presented time series of RR intervals (consecutive heartbeats) with less than 95% sinus beats were excluded from the study [18].

### 2.2. Ethical Aspects

All individuals were informed about the objectives and procedures of the study and, after agreeing to participate, signed a free and informed consent. All study procedures followed the Declaration of Helsinki and were approved by the Research Ethics Committee of the Faculty of Science and Technology, Presidente Prudente (approved on 4 October 2013, number 22530813.9.0000.5402). 

### 2.3. Data Collection

Data collection was performed in a room with temperature between 21 °C and 23 °C and humidity between 40% and 60%, in the afternoon between 13:00 and 18:00 to minimize the influences of circadian rhythm [19]. Prior to the evaluations, the volunteers were instructed not to drink alcoholic beverages and/or stimulants of the autonomic nervous system, such as coffee, tea, and chocolate, in the 24 h preceding the evaluation. In addition, the volunteers were also instructed not to engage in intense physical activity 8 h prior to data collection, to suspend the use of diuretics 24 h before the test, and not to drink liquids or consume food 4 h before the evaluations [20]. To guarantee reliability, the evaluators of this study were trained prior to data collection to ensure familiarity with all variables measured. The measures were carried out in 2014 and analyzed in 2017. 

### 2.4. Characterization of the Sample

Information about age, gender, disease duration, and drug therapy use was collected. In addition, body weight, stature, and casual blood glucose were measured. A digital scale (Welmy R/I 200, São Paulo, Brazil) was used to measure body mass, and a stadiometer for height measurement (Sanny, São Paulo, Brazil). From the data obtained, the body mass index (BMI) = weight/height^2^ (kg/m^2^) was calculated according to the Brazilian Guidelines for Obesity [21]. 

Casual blood glucose was measured by means of a digital pulp puncture test. For this analysis a drop of blood was deposited on OneTouch Ultra reagent tape (Johnson and Johnson Medical Brazil, São Paulo, Brazil) and analyzed by the OneTouch Ultra glucose meter (Johnson and Johnson Medical Brazil, São Paulo, Brazil) [22]. The volunteers were not restricted as to their feeding and fasting period, since the objective was only to establish the glycemic measures at the moment of data collection.

### 2.5. Assessment of Risk Factors

The risk factors evaluated in this study were blood pressure, resting heart rate, and fat percentage, as alterations in these factors are associated with an increased risk of CVD [15,23,24,25] as well as which, in subjects with diabetes, these factors are related to alterations in autonomic modulation [12,13,26,27,28]. 

Systolic blood pressure (SBP) and diastolic blood pressure (DBP) were measured indirectly using a stethoscope (Littman, Saint Paul, MN, USA) and aneroid sphygmomanometer (Welch Allyn-Tycos, New York, NY, USA) [29] on the left arm of the individual in the sitting position, using the criteria established by the VI Brazilian Guidelines for Hypertension [30]. Subjects with systolic blood pressure equal to or greater than 140 mmHg and/or diastolic blood pressure equal to or greater than 90 mmHg were considered as presenting high blood pressure [30]. Subjects with blood pressure values below 140/90 mmHg, but who reported using hypertension drugs, were also considered as a group with high blood pressure.

In order to verify resting heart rate, the volunteers were instructed to remain at rest for five minutes in the seated position, after which the heart rate (HR) was measured using the Polar S810i heart rate monitor, previously validated for heart rate recording [31] (Polar Electro, Kampele, Finland). Heart rate was divided, based on the median value, into two groups and subjects in the highest group for resting heart rate (≥88 beats/min) were classified as having a high heart rate and the subjects in the lowest group (≤87 beats/min) as a low heart rate.

Finally, the body fat percentage was obtained by means of the bio-impedance equipment Maltron BF 906 Body Fat Analyzer (Maltron, Rayleigh, Essex, UK) [32] with the individual in the supine position, on a nonconductive surface, without contact with metal, lower limbs abducted at 45º and upper limbs abducted at 30º to avoid contact of the limbs with the trunk. For the classification of body fat percentage, men with values equal to or greater than 30% body fat and women with values equal to or greater than 25% were classified as having high body fat [33].

### 2.6. Aggregation of Risk Factors

The aggregation of risk factors was determined by the sum of the presence of high blood pressure, high heart rate, and high fat percentage. The volunteers were stratified by groups of 0 risk factors, 1 risk factor, and 2 or more associated risk factors.

### 2.7. Autonomic Assessment

For autonomic evaluation, a catch strap was placed on the chest of the volunteers in the region of the distal third of the sternum and the Polar S810i heart rate receiver on the wrist (Polar Electro); equipment previously validated for recording heart rate beat-to-beat and for the use of the data for HRV analysis [31]. After placement of the brace and monitor, the volunteers were placed in the dorsal decubitus position and remained at rest for 30 min. The volunteers were instructed to remain resting, awake, spontaneously breathing, and avoid conversations during the assessment. 

For analysis of the HRV indices, heart rate was recorded beat-to-beat throughout the experimental protocol. From the period of greatest signal stability, 1000 consecutive RR intervals were selected after digital filtering by Polar Precision Performance SW software (version 4.01.029, Polar Electro Oy, Kempele, Oulu, Finland), complemented by manual filtering for elimination of premature ectopic beats and artifacts, and only series with more than 95% beats were included in the study [18].

For the analysis of HRV, linear indices in the time and frequency domains were used [34], as well as nonlinear indices that reflect the parasympathetic autonomic modulation. HRV analysis in the time domain was performed using the rMSSD (square root of the mean square of the differences between the adjacent normal RR intervals) and pNN50 indices (percentage of adjacent RR intervals with a duration difference greater than 50 ms). For analysis of HRV in the frequency domain, the high frequency spectral component (HF: 0.15–0.40 Hz) was used in a standardized unit and the spectral analysis was calculated using the fast Fourier transform algorithm. The nonlinear indices analyzed were the SD1 index (standard deviation of the instantaneous beat-to-beat variability) extracted from the Poincaré plot [34], and the 2LV (pattern with two similar variations) and 2ULV indices (pattern with two different variants) extracted from the symbolic analysis [35]. 

The rMSSD, pNN50, HF, and SD1 indices were calculated using Kubios HRV analysis software—version 2.0 [36](Kubios, Biosignal Analysis and Medical Image Group, Department of Physics, University of Kuopio, Kuopio, Finland), while the 2LV and 2ULV indices were calculated using software specific for nonlinear analysis [35].

### 2.8. Data Analysis

The normality of the data was verified by the Kolmogorov–Smirnov test. The study variables are presented as mean, standard deviation, proportion, and integers. The comparison between the mean values of the characterization data of the sample (age, disease duration, casual blood glucose, weight, height, BMI, systolic blood pressure, diastolic blood pressure, heart rate, and fat percentage), HRV indices (rMSSD, HF n.u, pNN50, SD1, 2LV, and 2ULV); and the sum of the risk factors (percentage fat, blood pressure, and resting heart rate) were performed by means of the analysis of variance (ANOVA) technique. Possible differences were identified by Bonferroni’s post-hoc test.

The relationship between the number of cardiovascular risk factors and the HRV indices was verified by means of binary logistic regression analysis in the unadjusted and adjusted models (considering the gender, age, BMI, obesity, disease duration, and glycemic control of the subjects). Statistical significance was set at 5% with a confidence interval of 95%. Data analysis was performed through the Statistical Package for the Social Sciences—version 15.0 (SPSS Inc., Chicago, IL, USA).

## 3. Results

Initially, data from 43 individuals with type 1 diabetes were analyzed. Of this total, four volunteers (9.3%) were excluded, as they presented an error in the RR intervals series greater than 5%. Thus, the final sample consisted of 39 young people with type 1 diabetes (19 men and 20 women, *p* = 0.384). The general characteristics of these volunteers are shown in Table 1. A statistically significant difference was observed in HR, fat percentage, height, BMI, and casual blood glucose among individuals with one or more associated risk factors when compared to those without risk factors. No statistically significant differences between the group with one risk factor and two or more risk factors were observed. 

Of the volunteers analyzed, three presented obesity (two grade II obesity and one grade III obesity) and the disease duration ranged from 3 to 26 years. While only three patients had three risk factors, all the individuals with two or more risk factors were included in the same group. In addition, all participants were insulin dependent and 20 (51.28%) used other drugs besides insulin (Appendix A).

In relation to the association between the parasympathetic indices of HRV and the aggregation of risk factors, as shown in Table 2, an inverse relationship can be observed in individuals with type 1 diabetes, i.e., the higher the number of aggregate risk factors. These individuals have lower chances of having higher rMSSD (16%), PNN50 (17%), and SD1 (21%) indices, regardless of gender, age, BMI, obesity, disease duration, and glycemic control of the subjects.

The comparison of values of the HRV indices reflecting parasympathetic modulation with the presence of cardiovascular risk factors studied is shown in Table 3. A significant difference was observed for the rMSSD and SD1 indices between individuals with type 1 diabetes who did not present cardiovascular risk factors and those with two aggregate risk factors. No statistically significant differences between the group with one risk factors and two or more risk factors were observed. 

## 4. Discussion

The main results of this study indicated that, independent of gender, age, BMI, obesity, disease duration, and glycemic control, individuals with type 1 diabetes mellitus with two aggregate cardiovascular risk factors presented a reduction in the values of the indices that reflect parasympathetic autonomic modulation compared to individuals who did not present the risk factors. In addition, it is possible to observe that the same pattern is repeated for the other indices, although without significant differences. Furthermore, higher values were observed for the measures of HR, fat percentage, height, BMI, and casual blood glucose in the young people with one or more associated risk factors in relation to those without risk factors.

Several studies have evaluated the behavior of the autonomic nervous system in individuals with diabetes in the presence of isolated risk factors [12,13,14,26], however, this is the first study to evaluate the aggregation of cardiovascular risk factors on the parasympathetic modulation of young people with a diagnosis of type 1 diabetes. In this study, we evaluated the presence and aggregation of elevated resting heart rate, resting blood pressure, and body fat, which are indicated by some authors as risk factors related to modifications in the autonomic behavior of individuals with diabetes [12,26,27].

The results found showed that, as observed in healthy individuals [1,2,3], the aggregation of risk factors for the development of CVD is also related to great loss in cardiac autonomic modulation in young people with type 1 diabetes. This finding should be considered by health professionals and organizations, since in individuals with diabetes there is a relationship between impairment in autonomic modulation and worse clinical prognosis [37], added to which when associated with risk factors, it may increase the chances of developing CVD and mortality [23,38]. In addition, it is worth noting that the parasympathetic nervous system is responsible for stimulating cells in the pancreas to release insulin [39,40], thus, in individuals with insulin production deficiency, such as type 1 diabetes, a dysfunction in this field may contribute to elevated blood glucose levels by altering the glycolytic metabolism.

Studies that analyzed the risk factors addressed in this study individually have demonstrated their relation with worse autonomic modulation [12,13,14]. Heart rate is regulated by the autonomic nervous system and its elevation reflects a predominance of the sympathetic branch over the parasympathetic. This autonomic imbalance is considered a risk factor for the development of CVD and can lead to fatal arrhythmias [16]. In a previous study, our group demonstrated that young people with type 1 diabetes presented high resting heart rate values associated with reduced parasympathetic modulation and overall variability [12], which was also observed in middle-aged adults with type 2 diabetes.

Increased resting heart rate is also related to a higher chance of developing autonomic neuropathy in older adults (60.42 ± 8.68 years) [26] and increased cardiovascular morbidity and mortality in middle-aged individuals (46 ± 6 years) with diabetes [41] and in older adults (median age 68 years) with coronary artery disease associated with diabetes [42].

In addition, the clinical condition of hypertension promotes changes in the sympathetic-vagal balance such as reduced parasympathetic tone and global autonomic control [43,44], including early dysregulation of the parasympathetic branch in newly diagnosed hypertensive individuals [44]. According to Voulgari et al. [13,45], there is a significant and positive association between the chance of cardiac autonomic neuropathy and several risk factors, including systolic blood pressure, for individuals with type 1 diabetes, showing that the odds of cardiac autonomic neuropathy increases with increases in systolic blood pressure. This result is consistent with our study because we showed that the autonomic function is reduced in individuals with two aggregate cardiovascular risk factors, among them the systolic blood pressure, reinforcing the important relation between risk factors and autonomic function in type 1 diabetes.

Another factor investigated in this study, related to changes in autonomic behavior, was the presence of elevated levels of body fat. Intra-abdominal adipose tissue is innervated by parasympathetic fibers and, in this way, the vagus nerve is responsible for modulating the endocrine function of adipose tissue, influencing the etiology of obesity [46]. A literature review indicates that a decrease in HRV is associated with a high percentage of body fat and a lower percentage of muscle mass [14] and this association may be at least partially mediated by insulin resistance [47]. In individuals with type 2 diabetes, autonomic dysfunction is associated with the presence of obesity [28].

The study limitations should be presented. Unfortunately, we could not evaluate data on HbA1c in four subjects (10% of total sample), however, it is already described in the literature that chronic hyperglycemia is associated with impairment in parasympathetic autonomic modulation [7,48]. Moreover, hypertension was defined based on one single value recorded at the time of autonomic testing, however volunteers with blood pressure values inferior to 140/90 mmHg who reported the use of hypertension drugs were considered with hypertension. In addition, the presence of obesity in three of the volunteers analyzed, as well as the variation in disease duration of type 1 diabetes, may have influenced the results obtained. As positive points, it stands out that this is the first study to evaluate the association between risk factors for CVD and parasympathetic indices in individuals with type 1 diabetes. Autonomic impairment in these individuals is already known in the literature, and the results of the present study demonstrate that the other factors, as well as the glycemic levels, should be investigated in this population.

Several studies showed that in the population with type 1 diabetes the presence of risk factors may contribute to the development of complications, including cardiovascular and autonomic complications [12,23,37,38], therefore, prevention and treatment strategies in this population become fundamental [49,50,51,52].

## 5. Conclusions

In conclusion, it has been found that young people with type 1 diabetes who present two aggregated risk factors have lower values in the marker indices of parasympathetic autonomic modulation compared to those without risk factors. In addition, our results show that regardless of the presence of factors such as sex, age, BMI, obesity, duration of disease, or glycemic control, the autonomic modulation parasympathetic remains altered in the presence of aggregation of cardiovascular risk factors. This information is important to understand that the aggregation of risk factors can accentuate autonomic alterations in this population, aggravating the state of health and clinical prognosis. Furthermore, it may help professionals working with this population to understand how risk factors could be related to autonomic alterations and to alert patients to possible complications and the importance of modifying daily behaviors and habits considered as risks for the development of CVD.

## Figures and Tables

**Table 1 medicina-55-00534-t001:** Characteristics of volunteers with type 1 diabetes, with zero, one, and two or more risk factors (n = 39).

**Variables**	**Zero (n = 14)**	**One (n = 15)**	**≥Two (n = 10)** *	*P*
		**Mean ± SD**		
Age (years)	22.28 ± 5.07	21.93 ± 2.96	23.80 ± 4.73	0.550
Disease duration	11.42 ± 6.69	10.46 ± 4.74	14.00 ± 6.56	0.353
Random blood glycemia (mg/dl)	153.64 ± 92.44	156.73 ± 87.50	243.8 ± 97.52 ^a^	0.042
Body weight (kg)	70.33 ± 13.20	75.40 ± 17.73	75.23 ± 14.98	0.632
Height (m)	1.76 ± 0.10	1.68 ± 0.09	1.66 ± 0.08 ^a^	0.028
BMI (Kg/m²)	22.51 ± 2.95	26.43 ± 5.70 ^a^	27.10 ± 3.96 ^a^	0.026
SBP (mmHg)	109.42 ± 9.99	110.00 ± 10.00	119.00 ± 15.95	0.113
DBP (mmHg)	61.14 ± 9.27	68.66 ± 11.87	68.66 ± 11.87	0.060
Pulse pressure	48.28 ± 9.50	41.33 ± 8.33	48.00 ± 13.16	0.122
HR (bpm)	73.14 ± 6.57	81.80 ± 8.51 ^a^	89.00 ± 10.21 ^a^	0.000
Body fat (%)	19.43 ± 6.35	27.66 ± 11.42 ^a^	31.73 ± 6.77 ^a^	0.004
**Risk Factors proportion**	**N (%)**			
	**One (n = 15)**		**≥Two (n = 10)**	
High SBP and/or DBP (mmHg)	2 (13.33%) 5 (33.33%) 8 (53.33%)		5 (50.00%) 8 (80.00%) 9 (90.00%)	
High HR (bpm)				
High body fat (%)				

^a^ Difference between zero cardiovascular risk factors (*p* < 0.05). Legend: SD = standard deviation; BMI = body mass index; SBP = systolic blood pressure; DBP = diastolic blood pressure; HR = heart rate; mg/dl = milligrams per deciliter; kg = kilogram; m = meter; mmHg = millimeter of mercury; bpm = beats per minute; % = percentage. * Three subjects had three risk factors.

**Table 2 medicina-55-00534-t002:** Association between aggregation of cardiovascular risk factors and linear indices of heart rate variability in volunteers with type 1 diabetes.

Cluster of Cardiovascular Risk Factors (≥2 Risk Factors)						
	Unadjusted			Adjusted *		
	Odds Ratio	CI (95%)	*P*	Odds Ratio	CI (95%)	*P*
rMSSD	0.93	0.86; 0.98	0.021	0.84	0.72; 0.99	0.042
HF n.u	0.97	0.92; 1.02	0.245	0.97	0.90; 1.03	0.356
pNN50	0.92	0.84; 0.99	0.048	0.83	0.71; 0.98	0.037
SD1	0.90	0.81; 0.98	0.021	0.79	0.63; 0.99	0.043
2LV	0.94	0.84; 1.05	0.301	0.94	0.77; 1.13	0.527
2ULV	0.97	0.87; 1.06	0.966	0.87	0.71; 1.06	0.874

Legend: rMSSD = root-mean square of differences between adjacent normal RR intervals in a time interval expressed in milliseconds; HF n.u = high frequency in normalized units; pNN50 = percentage of adjacent RR intervals with difference of duration greater than 50 ms; SD1 = standard deviation of the instantaneous variability beat to beat; 2LV = patterns with two like variations; 2ULV = patterns with two unlike variations. * adjusted by gender, age, BMI, obesity, disease duration, and glycemic control.

**Table 3 medicina-55-00534-t003:** Comparison of average values of indices of heart rate variability according to the sum of cardiovascular risk factors in volunteers with type 1 diabetes.

Number of CRF	Zero (n = 14)	One (n = 15)	≥Two (n = 10)	*P*
Mean ± SD				
rMSSD	38.82 ± 15.60	35.02 ± 18.43	20.75 ± 13.32 ^a^	0.031
HF n.u	47.20 ± 9.97	46.98 ± 14.71	40.72 ± 20.37	0.515
pNN50	19.53 ± 15.08	16.52 ± 17.59	5.57 ± 9.68	0.082
SD1	27.47 ± 11.03	24.78 ± 13.04	14.70 ± 9.52 ^a^	0.031
2LV	13.64 ± 7.34	11.22 ± 4.77	9.83 ± 8.18	0.374
2ULV	18.52 ± 9.51	17.66 ± 9.89	15.77 ± 5.98	0.756

Legend: CRF = cardiovascular risk factors; SD = standard deviation; rMSSD = root-mean square of differences between adjacent normal RR intervals in a time interval expressed in milliseconds; HF n.u = high frequency in normalized units; pNN50 = percentage of adjacent RR intervals with difference of duration greater than 50ms; SD1 = standard deviation of the instantaneous variability beat to beat; 2LV = patterns with two like variations; 2ULV = patterns with two unlike variations. The values in bold represent *p* < 0.05; ^a^ difference between individuals with type 1 diabetes who did not present cardiovascular risk factors and those with two aggregate risk factors.

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
