# Peer review of "Relationship of the Aggregation of Cardiovascular Risk Factors in the Parasympathetic Modulation of Young People with Type 1 Diabetes"

_medicina, 2019, doi:10.3390/medicina55090534_

Round 1
Reviewer 1 Report
This is a small clinical study investigating the aggregation of cardiovascular risk factors in the parasympathetic modulation of people with type 1 diabetes. Below are my comments to the authors
1. Introduction: You could improve the rationale for the reader on why this study was done
2. Methods: The assessment of the risk factors belongs to methods, not in the results section.
3. There is poor justification of why the specific (three) risk factors were chosen.
4. Other important cardiovascular risk factors were not included in this study such as obesity, smoking, no exercise, dyslipidemia, poor glycemic control, duration of diabetes. Why didn’t the authors include these variables in the analysis? This needs to be addressed. If the focus is only these three factors then along with justification there needs to be adjustment for other risk factors (random glucose/obesity/duration of diabetes etc) in the linear regression model.
5. Body muscle mass is not presented in the results, why is it in the methods? Is the use of electronic scale a validated method for measuring body muscle mass?
6. One of the main three risk factors is high resting heart rate defined as </= 88 beats/minute. This does not make sense.
7. The definition of resting heart rate based on quartiles from a relatively small size is a poor selection criterion. Instead, authors could use validated criteria to define resting heart rate. Furthermore, resting heart rate can be varied by chronic exercise, age, medication use, which are not adjusted factors in the regression.
8. Discussion: The hypertension promotes changes in sympathetic –vagal balance (line 250) requires more explanation.
Author Response
Anne Kastelianne França da Silva
PhD Candidate
Stress Physiology Laboratory
17thJune 2019
Roberto Simonsen Street, 305
Post Code 19060-900
Presidente Prudente, SP
BRAZIL
Telephone: +55 18 32295819
Email: anne_franca@hotmailc.com
Dear editors and reviewers of Medicina Journal,
On behalf of all the authors, I would like to thank the opportunity to revise our manuscript entitled “Relationship of the aggregation of cardiovascular risk factors in the parasympathetic modulation of young people with type 1 Diabetes”, byAnne Kastelianne França da Silva, Diego Giuliano Destro Christofaro, Laís Manata Vanzella, Franciele Marques Vanderlei, Maria Júlia Lopez Laurino and Luiz Carlos Marques Vanderlei.
We also took the opportunity to thank the reviewers by valuable comments. We are forwarding the answer point-to-point to modifications requested by the reviewers. The manuscript was modified according with the suggestions, and the revised version is attached. We look forward to hearing from you.
Anne Kastelianne França da Silva
São Paulo State University (UNESP)
Reviewer #1
Initially, we thank the reviewer for their comments in this first review that certainly contributed to improving the quality and clarity of the manuscript. We consider all suggestions/comments appropriate and we are sending the answer point-to-point of questions raised by the reviewer. The changes in the revised manuscript are indicated in yellow highlight.
1. Referee's comment: Introduction: You could improve the rationale for the reader on why this study was done
Answer: We thank you for the suggestion. This information was reinforced in the text [Line 61-66].
2. Referee's comment:Methods: The assessment of the risk factors belongs to methods, not in the results section
Answer: The information was inserted on topic “2.6. Aggregation of risk factors” at Methods section [Line 135-136].
3. Referee's comment: There is poor justification of why the specific (three) risk factors were chosen
Answer: We understood the concern of the reviewer, which was also ours when defining which risk factors we would insert in the study. Two aspects guided this choice: 1) The risk factors included in our study are among the main modifiable risk factors for CVD and autonomic dysfunction in diabetes; 2) These factors represent variables that can be easily measured in clinical practice. Blood pressure is one of the three most important risk factors for CVD in diabetic population, in addition, the glycated hemoglobin and cholesterol1, and can be easily measured by any health professional. Factors related to body composition, such as obesity and diet, are also important in these individuals1. Therefore, we chose to include fat percentage as a measure, because we believe it to be a better predictor of body composition than BMI measurements. In addition, as already demonstrated in other studies, HR is associated with autonomic dysfunction in the diabetic population2. This information has been reinforced in the text [Line 57-61] and as requested by the reviewer the linear regression model was adjusted by the other risk factors that we had available (BMI, obesity, disease duration and glycemic control). We request the understanding of the reviewer to maintain the analysis of how it was done.
1-Maahs DM, Daniels SR, De Ferranti SD, et al. Cardiovascular disease risk factors in youth with diabetes mellitus: A scientific statement from the American heart association. Circulation 2014; 130: 1532–1558.
2-Silva AKF, Christofaro DGD, Vanderlei FM, et al. Association of cardiac autonomic modulation with physical and clinical features of young people with type 1 diabetes. Cardiol Young2016; 1–9.
4. Referee's comment: Other important cardiovascular risk factors were not included in this study such as obesity, smoking, no exercise, dyslipidemia, poor glycemic control, duration of diabetes. Why didn’t the authors include these variables in the analysis? This needs to be addressed. If the focus is only these three factors then along with justification there needs to be adjustment for other risk factors (random glucose/obesity/duration of diabetes etc) in the linear regression model.
Answer: Dear reviewer thank you for the comment. As suggest the new adjusted on statistical analysis considered BMI, obesity, disease duration and glycemic control (blood glucose) in the linear regression model [Page 6; Table 3]
5. Referee's comment: Body muscle mass is not presented in the results, why is it in the methods? Is the use of electronic scale a validated method for measuring body muscle mass?
Answer: Dear reviewer again thank you for the comment. We performed the assessment of the body weight and not body muscle mass. The word correct is “weight” and not “mass”. The correct word was insert [Methods: Line 100].
6. Referee's comment: One of the main three risk factors is high resting heart rate defined as </= 88 beats/minute. This does not make sense.
Answer: Dear reviewer, thank you for your note. This information was adjusted [Line 125].
7. Referee's comment: The definition of resting heart rate based on quartiles from a relatively small size is a poor selection criterion. Instead, authors could use validated criteria to define resting heart rate. Furthermore, resting heart rate can be varied by chronic exercise, age, medication use, which are not adjusted factors in the regression.
Answer: Dear reviewer, we understand the concern with the definition of resting heart rate in our sample and agree that it is a variable that can be influenced by several factors. The use a valid criterion to define resting heart rate is difficult, and so, we chose to classify the sample with reference to the group median to stratify the sample in high / low resting heart rate. Regarding adjustments in the regression model, we believe that because of similar age between groups and use of medications without a direct influence on heart rate control, adjustments would not be necessary.
8. Referee's comment: Discussion: The hypertension promotes changes in sympathetic –vagal balance (line 250) requires more explanation.
Answer: The information was better explained [Line 263].

Reviewer 2 Report
The aim of the study of Anne Kastelianne França da Silva and colleagues is very interesting and highlight relevant information regarding risk factor of parasympathetic modulation in young people with type 1 Diabetes.
However, major significant issues arose reading the manuscript and for this reason, the manuscript needs major revision prior to a publication:
Major revisions:
1) English revision is needed (at the end some suggestions)
2) Sample size calculation: what is the primary aim of the study? Reading the paper the primary aim seems to be the evaluation of differences in the indices of heart rate variability in patients with different risk factors (0, 1 and ≥2), for this reason, you should calculate the sample size for an ANOVA test and not for the correlation.
3) HbA1c: data about HbA1c of patients should be collected; data from clinical appointments 3 months previous or after the clinical assessment could be used. If it is impossible to collect these data, statistical system to evaluate the missing values should be used.
4) Results should be adjusted for disease duration, glycaemic control (blood glucose and/or HbA1c), BMI and obesity, at least.
5) Information about microvascular disease should be collected and evaluated among the results. In particular presence of CAN should be considered and related to the parasympathetic modulation.
Minor revisions:
1) Add graphs for significant correlation in table 3
2) line 46: which risk factors are you talking about?
3) line 58: specify which cardiovascular risk factor do you want to take into account.
4) line 69-72: consider revision highlighting exclusion and inclusion criteria. "cardiorespiratory diseases" should be specified. Medical history of myocardial infarction and other heart diseases should be added among the exclusion criteria (considered the age of the patients, it should not be a problem).
5) line 116: check HR values and describes also the thresholds for the other quartiles.
6) line 129: what is the meaning of "after the initial orientations"?
7) Analysis in Table 1 and 2: there is any significant difference between patients with 1 risk factor and patients with 2 risk factors? Specify if there is no difference.
8) Table 1: "high SBP and/or SBP" do you mean DBP?
9) Bibliography: consider only references in English or both English and other languages, not other languages alone.
Suggestions:
1) line 352: "newly diagnosed 251 individuals" consider specifying of which disease.
2) line 245: explain in more details the findings in type 1 diabetes [REF 23 and 43] and if they are consistent whit the results of the study
3) line 244: ref 38 could be omitted
English revisions:
line 39: word “tool”, consider revision
line 39-43: consider English revision
line 43: use always the same verbal tense throughout the manuscript
line 54: "is subject" consider revision
line 64: consider change "diagnosis time" with disease duration, throughout the manuscript
line 79: What do you mean with the words "all collections"? Consider revision, for example, all measurements/tests, etc
line 90: consider English revision, eg "information about age, gender [...] was collected."
line 104: consider revision of "in person with" eg in subjects
line 125: consider to delete "or absence"
line 135: consider revise collection with test
line 135: the sentence "After collection of the autonomic modulation the volunteers were released." could be deleted
line 158-164: consider English revision
line 176: delete the word variables
line 228-233: consider English revision
line 229: loss not losses
LINE 231: "organs" consider revision eg "organization"
line 238: "in isolation" consider revision
line 268: “bellow” consider correction
line 269: "with hypertensive" consider revision
line 271-275: consider English revision
line 276-279: consider English revision
line 280: consider revision eg: "In conclusion, it has been found that young people with type 1 diabetes [...]"
line 283-288: consider English revision
point out: too much
In Chart 2 consider changing the word "medicine" with therapy
Author Response
Anne Kastelianne França da Silva
PhD Candidate
Stress Physiology Laboratory
17thJune 2019
Roberto Simonsen Street, 305
Post Code 19060-900
Presidente Prudente, SP
BRAZIL
Telephone: +55 18 32295819
Email: anne_franca@hotmailc.com
Dear editors and reviewers of Medicina Journal,
On behalf of all the authors, I would like to thank the opportunity to revise our manuscript entitled “Relationship of the aggregation of cardiovascular risk factors in the parasympathetic modulation of young people with type 1 Diabetes”, byAnne Kastelianne França da Silva, Diego Giuliano Destro Christofaro, Laís Manata Vanzella, Franciele Marques Vanderlei, Maria Júlia Lopez Laurino and Luiz Carlos Marques Vanderlei.
We also took the opportunity to thank the reviewers by valuable comments. We are forwarding the answer point-to-point to modifications requested by the reviewers. The manuscript was modified according with the suggestions, and the revised version is attached. We look forward to hearing from you.
Anne Kastelianne França da Silva
São Paulo State University (UNESP)
Reviewer #2
Initially we thank the reviewer for their comments in this first review that certainly contributed to improving the quality and clarity of the manuscript. We consider all suggestions/comments appropriate and we are sending the answer point-to-point of questions raised by the reviewer. The changes in the revised manuscript are indicated in yellow highlight.
1. Referee's comment: English revision is needed (at the end some suggestions).
Answer: We thank the reviewer for the comment and suggestions. The manuscript was translated by a native in language (Dra. Robin Camargo – Nationality: British; Qualifications: ACCA Certificate Stage, AAT, 3 A’Levels and 9 O’Levels) and as suggested was revised again in an attempt to improve the grammar of English.
2. Referee's comment: Sample size calculation: what is the primary aim of the study? Reading the paper the primary aim seems to be the evaluation of differences in the indices of heart rate variability in patients with different risk factors (0, 1 and ≥2), for this reason, you should calculate the sample size for an ANOVA test and not for the correlation.
Answer: The primary aim of the study is the relationship between the indices of heart rate variability and aggregation of risk factors. This information was better clarify in the paragraph about the objective [Line 63-66].
3. Referee's comment: HbA1c: data about HbA1c of patients should be collected; data from clinical appointments 3 months previous or after the clinical assessment could be used. If it is impossible to collect these data, statistical system to evaluate the missing values should be used.
Answer: Dear reviewer we agree with this comment, fortunately, we can evaluate data on HbA1c in only 4 subjects (10% oftotal sample). However, we considered this an important limitation and was insert in the end of discussion [Line 276-277].
4. Referee's comment: Results should be adjusted for disease duration, glycemic control (blood glucose and/or HbA1c), BMI and obesity, at least.
Answer: Dear reviewer thank you for the comment. As suggest the new adjusted on statistical analysis considered BMI, obesity, disease duration and glycemic control (blood glucose) in the liner regression model [Page 6; Table 3].
5. Referee's comment: Add graphs for significant correlation in table 3
Answer: Based on the reviewer comment, the graph of significant correlation was inserted in the results [Page 6; Figure 1].
6. Referee's comment: line 46: which risk factors are you talking about?
Answer: We are talking about the risk factors studied by Colhum et al. and the EURODIAB Prospective Complications Study Group [Line 46-52].
7. Referee's comment: line 58: specify which cardiovascular risk factor do you want to take into account.
Answer: This information was insert [Line 63-64].
8. Referee's comment: line 69-72: consider revision highlighting exclusion and inclusion criteria. "cardiorespiratory diseases" should be specified. Medical history of myocardial infarction and other heart diseases should be added among the exclusion criteria (considered the age of the patients, it should not be a problem).
Answer: This information was specified [Line 78-79].
9. Referee's comment: line 116: check HR values and describes also the thresholds for the other quartiles.
Answer: This information was checked and inserted [Line 122].
10. Referee's comment:line 129: what is the meaning of "after the initial orientations"?
Answer: The aim was only to emphasize that the data collection was performed after the orientations about “to remain resting, awake, spontaneously breathing, avoiding conversations during collection” [Line 140-141]. How this phrase can confuse the reader, we exclude from the text.
11. Referee's comment: Analysis in Table 1 and 2: there is any significant difference between patients with 1 risk factor and patients with 2 risk factors? Specify if there is no difference.
Answer: The analyze between the group with 1 risk factor and 2 risk factors were performed and not showed difference between them. This information was inserted [Line 193-194] and the legend the table was adjusted.
12. Referee's comment: Table 1: "high SBP and/or SBP" do you mean DBP?
Answer: We would like to thank you for the question. The DBP was assessment and insert in table 1 [Page 5].
13. Referee's comment: Bibliography: consider only references in English or both English and other languages, not other languages alone
Answer: The references were adjusted and highlighted in yellow.
Suggestions:
1. line 352: "newly diagnosed 251 individuals" consider specifying of which disease.
Answer: Thank you for the suggestion. The condition was inserted.
2. line 245: explain in more details the findings in type 1 diabetes [REF 23 and 43] and if they are consistent whit the results of the study
Answer: This question was better explained on paragraph [line 273-277].
3. line 244: ref 38 could be omitted
Answer: The reference was omitted
4. English revisions:
Answer: Thank you again for the suggestions. The English was revised by a native in language English (Dra. Robin Camargo – Nationality: British; Qualifications: ACCA Certificate Stage, AAT, 3 A’Levels and 9 O’Levels).

Round 2
Reviewer 1 Report
No more comments
Author Response
Anne Kastelianne França da Silva
PhD Candidate
Stress Physiology Laboratory
11thJuly 2019
Roberto Simonsen Street, 305
Post Code 19060-900
Presidente Prudente, SP
BRAZIL
Telephone: +55 18 32295819
Email: anne_franca@hotmailc.com
Dear editors and reviewer of Medicina Journal,
On behalf of all the authors, I would like to thank the opportunity to revise our manuscript entitled “Relationship of the aggregation of cardiovascular risk factors in the parasympathetic modulation of young people with type 1 Diabetes”, by Anne Kastelianne França da Silva, Diego Giuliano Destro Christofaro, Laís Manata Vanzella, Franciele Marques Vanderlei, Maria Júlia Lopez Laurino and Luiz Carlos Marques Vanderlei.
We also took the opportunity to thank the reviewers by valuable comments. We are forwarding the answer point-to-point to modifications requested by the reviewers. The manuscript was modified according with the suggestions, and the revised version is attached. We look forward to hearing from you.
Anne Kastelianne França da Silva
São Paulo State University (UNESP)
Reviewer #1
We thank the reviewer for their comments and suggestions in the first review and we are glad to know that the modifications made based on their suggestions improved the quality and clarity of the manuscript. And again, we thank the reviewer for this second feedback.

Reviewer 2 Report
line 62 - consider change DCV in CVD
line 123 - two quartiles,
If you divide the population in quartiles you will obtain four groups, then you could choose the highest and the lowest. If I have understood correctly your work you have simply divide the population into two groups based on the HR value. So you should not use the word "quartile".
In this case, why have you chosen 88? You should provide a brief explanation
line 184 & 200: please change into: "No statistically significant differences between the group with one risk factors and =two risk factors were observed"
If the primary aim of your study is: "relationship between the aggregation of CVD risk 63 factors (blood pressure, resting heart rate and fat percentage) and parasympathetic indices of heart 64 rate variability in young people with type 1 diabetes." You should show these results before the differences between the groups. Please change accordingly.
Author Response
Anne Kastelianne França da Silva PhD Candidate Stress Physiology Laboratory 09th July 2019 Roberto Simonsen Street, 305 Post Code 19060-900 Presidente Prudente, SP BRAZIL Telephone: +55 18 32295819 Email: anne_franca@hotmailc.com Dear editors and reviewer of Medicina Journal, On behalf of all the authors, I would once again like to thank the opportunity to revise our manuscript entitled “Relationship of the aggregation of cardiovascular risk factors in the parasympathetic modulation of young people with type 1 Diabetes”, by Anne Kastelianne França da Silva, Diego Giuliano Destro Christofaro, Laís Manata Vanzella, Franciele Marques Vanderlei, Maria Júlia Lopez Laurino and Luiz Carlos Marques Vanderlei. We also took the opportunity to again thank the reviewers by valuable comments. We are forwarding the answer point-to-point to modifications requested by the reviewers. The manuscript was modified according with the suggestions, and the revised version is attached. We look forward to hearing from you. Anne Kastelianne França da Silva São Paulo State University (UNESP) Reviewer #2 Initially we thank the reviewer for their comments in this second review that certainly contributed to improving the quality and clarity of the manuscript. We consider all suggestions/comments appropriate and we are sending the answer point-to-point of questions raised by the reviewer. The changes in the revised manuscript are indicated in yellow highlight. Referee's comment: line 62 - consider change DCV in CVD Answer: Change made at line 61, the abbreviation in line 62 was correct. Referee's comment: line 123 - two quartiles, If you divide the population in quartiles you will obtain four groups, then you could choose the highest and the lowest. If I have understood correctly your work you have simply divide the population into two groups based on the HR value. So you should not use the word "quartile". In this case, why have you chosen 88? You should provide a brief explanation Answer: Indeed, as the reviewer rightly remarks, we divided the population into two groups. This was done based on the median of resting heart rate. So, we used 88 beats per minute as cut point because this was the value of the median. In order to make the text correct and clearer we changed the word “quartiles” to “groups” in lines 121, 122 and 123. Also, in lines 121 and 122 we added the following highlighted information: “Heart rate was divided, based on the median value, into two groups (…)” Referee's comment: line 184 & 200: please change into: "No statistically significant differences between the group with one risk factors and =two risk factors were observed" If the primary aim of your study is: "relationship between the aggregation of CVD risk 63 factors (blood pressure, resting heart rate and fat percentage) and parasympathetic indices of heart 64 rate variability in young people with type 1 diabetes." You should show these results before the differences between the groups. Please change accordingly. Answer: As requested the phrase “Differences between the group with one risk factors and the other groups (zero and ≥two) were not observed” was replaced by “No statistically significant differences between the group with one risk factors and ≥two risk factors were observed”. (Lines 212 and 213) Regarding the results presentation, we changed the order accordingly to the primary aim of the study, so the paragraph included in lines 191 to 195, describing the differences between groups, was moved along with Table 2 and placed after the Figure 1, starting now at line 209. Therefore, the order of the tables had to be adjusted in the text, so the former Table 2 (Comparison of average values of indices of heart rate variability according to the sum of cardiovascular risk factors in volunteers with type 1 Diabetes) is now named Table 3 and the former Table 3 is now named Table 2.
